## Research Article

conservation; biodiversity; endemism; threat; habitat loss

**Corresponding author:**
Jorge Antonio Gómez Díaz;
Email: jorggomez@uv.mx

# Assessing the extinction risk of Veracruz cycads

Jorge Antonio Gómez Díaz 🄳

Instituto de Investigaciones Biológicas, Universidad Veracruzana, Xalapa, Mexico and Centro de Investigaciones Tropicales, Universidad Veracruzana, Xalapa, Mexico

## Abstract

Cycads, an ancient lineage, face a higher threat of extinction than any other plant group. To address this urgent issue, a more comprehensive method for assessing extinction threat, the Conservation and Prioritization Index (CPI), is proposed and tested for cycads in the State of Veracruz, Mexico. The CPI is a multifaceted approach that incorporates techniques used in conservation status assessments by the IUCN and the Mexican NOM-059-SEMARNAT-2010 but incorporates other information, including georeferenced distribution data, endemism in Veracruz, number of locations, extent of occurrence, and distribution area. Using CPI, correlations were found between longitude and extinction risk for *Ceratozamia* species in Veracruz. *Zamia vazquezii* and *Z. inermis* were assessed to have the highest level of extinction risk. Overall, this study indicates that a more holistic approach, incorporating broader sources of environmental health, can be used to more effectively and proactively manage extinction threats to cycads in Veracruz. In this sense, Veracruz can serve as a model for conservation planning in different states in Mexico and worldwide. CPI is a tool that can be applied to other regions to manage another threatened biota. This method enhances objectivity and effectiveness in conservation efforts, promoting data-driven decision-making that can be used globally.

## Impact statement

The research presented in this article holds implications for both local and international conservation efforts. The study provides insights by assessing the extinction risk of cycad species in Veracruz. The impact of this research is found at various levels. Locally, in Veracruz, where cycad species richness is substantial, the findings highlight specific threats, such as habitat loss driven by human activities. This knowledge will inform regional conservation policies and strategies, ensuring the protection and preservation of these ecologically significant plants. At the national level, the study contributes to Mexico's broader understanding of cycad biodiversity and the associated risks, supporting the development of conservation initiatives on a larger scale. Internationally, the research adds information to the global discourse on plant conservation, especially concerning ancient gymnosperms like cycads. The identified knowledge gaps and the proposed conservation measures serve as a blueprint for addressing similar challenges cycad populations face in other regions worldwide. By emphasising the urgent need for continued research and monitoring, the study advocates for a proactive approach to biodiversity preservation, aligning with international goals for sustainable development. Overall, the impact of this research extends beyond academic circles, offering insights that can drive conservation outcomes at local, national, and global levels. The audience, including scientists, conservationists, policymakers, and stakeholders, plays a role in this process, and their participation is key to the success of these conservation efforts.

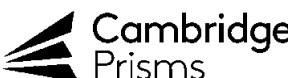



## Introduction

Cycads are an ancient lineage of gymnosperm plants that hold significant ecological and evolutionary importance (Wang and Ran, 2014). However, recent genetic studies show that extant species have radiated and diversified relatively recently, particularly from a few ancestral species during the late Miocene (Crisp and Cook, 2011; Nagalingum et al., 2011). Nonetheless, these organisms now confront a critical challenge in the face of global and regional threats (Donaldson, 2003). Cycads have captivated scientists and nature enthusiasts alike, serving as evolutionary models that provide insights into the history of plants (Salzman, 2019). Cycads occupy a distinct ecological niche, playing crucial roles in diverse ecosystems (Calonje et al., 2022). Despite their environmental importance and evolutionary heritage, cycads face various global and regional threats that jeopardise their survival (Yessoufou et al., 2017). At a global level, three prominent hotspots of cycad diversity are recognised. Oceania, particularly Australia, leads with the greatest diversity. Mesoamerica, with Mexico as its foremost representative, ranks second. Africa, specifically South Africa, holds the third position in the global cycad species count (Yessoufou et al., 2017; Calonje et al., 2023).

Mexico, with 74 species, is a crucial epicentre for the Zamiaceae family in the Neotropics (Calonje et al., 2023). The magnitude of this diversity is underscored by the presence of 37 species in the *Ceratozamia* genus, 20 in *Zamia*, and 17 in *Dioon* (Calonje et al., 2023). The extraordinary number of species, coupled with the fact that 88.9% of them are endemic, highlights the uniqueness and vulnerability of the cycad species in Mexico (Nicolalde-Morejón et al., 2014). However, the looming threat of extinction of all three genera emphasises the urgent need to address the protection and preservation of these species (Nicolalde-Morejón et al., 2014). The classification of most Mexican cycads under some risk category according to NOM-059-SEMARNAT-2010 (NOM-059) underscores the imminent need for effective conservation strategies. The continually expanding knowledge of cycad systematics, evolutionary biology, and conservation in Mexico not only enriches our understanding of current diversity but also provides essential tools to protect and safeguard the future of these plants on a global scale (Nicolalde-Morejón et al., 2014).

Veracruz ranks third in Mexico for cycad species richness, with 21 species identified across three genera: *Ceratozamia* (13 species), *Zamia* (6 species), and *Dioon* (2 species) (Calonje et al., 2023). Despite this diversity, the conservation of these species is increasingly threatened by habitat loss, driven primarily by human activities such as deforestation and urbanisation (Ellis et al., 2011). Data from the Mexican National Forestry Commission (CONAFOR, by its Spanish acronym) highlights a troubling pattern of deforestation in Veracruz, with peaks such as the 16,023 ha lost in 2010 (CONAFOR, 2023). This ongoing habitat degradation presents a serious obstacle to cycad conservation efforts, emphasising the urgent need to curb deforestation and protect the remaining natural environments in the state. Additionally, climate change, invasive species, and overexploitation exacerbate their challenges. Therefore, it becomes increasingly crucial to study and understand the vulnerabilities of cycad species in Veracruz to develop effective conservation strategies.

An important concept to consider when assessing the extinction risk of cycads is 'extinction debt.' Cronk (2016) highlighted the prolonged lag times experienced by long-lived plants, such as cycads, with extinction processes sometimes unfolding over several centuries. This slow progression towards extinction, often referred to as a 'slow-burning fuse,' is particularly relevant for cycads, which may face extinction risk long before the process becomes evident. Consequently, timely conservation actions during this critical window are essential to prevent the completion of the extinction process. However, a cautionary approach to conservation interventions is equally important, as improper actions could accelerate extinction. Therefore, this article aims to evaluate the extinction risks of Veracruz cycads. By synthesising available data and enhancing our understanding, I seek to inform effective conservation measures to protect these plants and their ecosystems.

## Methodology

### Study area

Veracruz is a diverse and ecologically significant region in eastern Mexico. It is in the east of Mexico, stretching from 17°10'N to 22°28'N latitude and 93°35'W to 98°39'W longitude. The Gulf of Mexico borders it at the east and encompasses a variety of ecosystems, including tropical rainforests, cloud forests, and coastal habitats. The complex topography, ranging from coastal plains to mountainous terrains reaching over 5,600 meters above sea level

(Pico de Orizaba), contributes to the environmental heterogeneity of Veracruz (Cruz-Angón, 2011). The state is characterised by its high biodiversity (Gómez-Pompa and Castillo-Campos, 2010). Veracruz has numerous Protected Areas (PA) and nature reserves. However, the current network of PA is deemed insufficient for safeguarding the state's biodiversity (Gómez Díaz et al., 2023). This inadequacy arises from the isolation of these areas and the absence of a systematically prioritised conservation scheme. Additionally, the current design must consider flagship and threatened species, such as cycads, further compromising conservation efforts.

### Extinction risk assessment

To assess the extinction risk of cycads in Veracruz, I used various information sources, including the IUCN (2023) conservation status assessments. While widely used, this assessment lacks certain complementary variables. Only one species, *C. dominguezii*, remains unevaluated due to its recent description (Pérez-Farrera et al., 2021). I conducted a preliminary assessment of this species using the *IUCN.eval* function from the *conR* package (Dauby et al., 2017) with records of the Global Biodiversity Information Facility (GBIF) corroborated by the species author (GBIF, 2023). To ensure data reliability, GBIF records were scrutinised for metadata – including data sources, collection methods, and georeferenced points. Taxonomic identifications were verified against original species descriptions, minimising biases.

Endemism to Veracruz was included using data from The World List of Cycads (Calonje et al., 2023). I incorporated risk status according to Mexican law NOM-059-SEMARNAT-2010 (NOM-059; SEMARNAT, 2010), which identifies at-risk species in Mexico but does not fully align with IUCN criteria. Since only 15 cycad species from Veracruz have been assessed under NOM-059, I evaluated the remaining six (*C. delucana*, *C. dominguezii*, *C. subroseophylla*, *C. tenuis*, *C. totonacorum*, and *Z. splendens*) using the MER method established by the law, utilising all available information. Although the MER method considers population genetics, it is flexible for species lacking such data, allowing inferences based on available indirect criteria.

I added information on the number of localities for each species from the IUCN. Four species (*C. dominguezii*, *C. euryphyllidia*, *C. mexicana*, and *Z. purpurea*) lacked sufficient data in the IUCN assessment. To address this gap, I used presence records from GBIF (2023), thoroughly reviewed for data quality. I enriched the dataset with herbarium records and consulted experts in Veracruz Zamiaceae (see Acknowledgements). Where possible, field verification confirmed the accuracy of the georeferenced data and species identifications. This combined information was used to calculate localities and apply the IUCN's Criterion B using the *IUCN.eval* function in the *conR* package.

I used the Extent of Occurrence (EOO) data from the IUCN assessments, except for *C. dominguezii*, for which I calculated it using the *EOO.computing* function in the *conR* package. I determined the percentage of each species distribution area within Veracruz using IUCN distribution maps (minimum convex polygon method). For nine species without existing maps (*C. brevifrons*, *C. delucana*, *C. dominguezii*, *C. subroseophylla*, *C. tenuis*, *C. totonacorum*, *Z. inermis*, *Z. splendens*, and *Z. vazquezii*), I created them using *EOO.computing* with *method.range* = "*convex.hull*" in *conR*.

I calculated the percentage of suitable habitat loss for each species using the Global Canopy Height dataset for the year 2019 (Potapov et al., 2021), with 30 m spatial resolution from GEDI lidar data. Canopy height thresholds were set based on species's habitat

preferences: >15 m for *Ceratozamia* species (associated with mature forests and mountain cloud forests; Carvajal-Hernández and Gómez-Díaz, 2024), > 5 m for *Dioon* species (adapted to open environments and low deciduous and dry forests; Vovides, 1990), >10 m for *Zamia* species (preferring medium sub-deciduous forests; González-Astorga et al., 2006), and >3 m for *Z. furfuracea* (inhabiting coastal dunes; Favian-Vega et al., 2022).

I calculated the percentage of each species' area outside PA by using species distribution polygons and layers representing federal and state PAs, as well as areas voluntarily dedicated to conservation (AVDC; CONABIO, 2023). Finally, I determined the number of municipalities in Veracruz where each species occurs by overlaying species distribution polygons with Mexico's municipal boundaries (Table 1). The variables included in the analysis were selected based on their relevance to assessing extinction risk and their alignment with established conservation frameworks, such as the IUCN Red List and Mexican NOM-059 criteria. Key factors like the number of localities, EOO, and habitat loss were chosen because they directly reflect population fragmentation, geographic range, and anthropogenic threats – critical components of species vulnerability (Murray et al., 2017). Endemism and distribution within Veracruz were included to prioritize regionally significant species, while the percentage of habitat outside PAs highlights conservation gaps (Gómez Díaz et al., 2023).

I converted categorical variables into numerical values based on risk levels. IUCN categories were scored as 4 for CR (Critically Endangered), 3 for EN (Endangered), 2 for VU (Vulnerable), and 1 for NT (Near Threatened). The scores for the two NOM-059 categories assigned to species were 2 for P (In danger of extinction) and 1 for A (Threatened). All variables were normalised to range between 1 and 0 using the formula (James et al., 2021):

$$z_i = \frac{x_i - (x)}{(x) - \min(x)}$$

where '$z_i$' is the normalized value, '$x_i$' is the original value, '$\min(x)$' is the minimum, and '$\max(x)$' is the maximum in the dataset. To assess extinction, standardised variables like the number of localities, EOO, and municipalities were inverted to assign higher risk values to lower original values (e.g., fewer localities or smaller EOOs). This is achieved using the following formula:

$$y_i = 1 - z_i$$

where '$y_i$' is the *i*th inverse normalised value, and '$z_i$' is the *i*th normalised value. This transformation inverted the scale of the normalised data.

Correlation among all these variables was analysed using Pearson's correlation coefficient. Subsequently, species were ordered using all the previous variables, except the EOO variable, due to its high correlation with the number of locations, municipalities, and IUCN, through a Multiple Factor Analysis (MFA). This method

**Table 1.** Raw data used to calculate the Conservation Prioritization Index (CPI) for cycad species in Veracruz, Mexico. The variables include IUCN (conservation category according to the IUCN Red List), Endemic (whether the species is endemic to Veracruz: Yes/No), NOM-059 (risk category under Mexican law NOM-059-SEMARNAT-2010, Loc (number of recorded localities), EOO (Extent of Occurrence in km²), Area (percentage of the species' distribution within Veracruz), Habitat Loss (percentage of habitat loss for the species), Out PA (percentage of the species' distribution outside Protected Areas), and Mun (number of municipalities in Veracruz where the species are found)

| Species | IUCN | Endemic | NOM-059 | Loc | EOO | Area | Habitat loss | Out PA | Mun |
|---|---|---|---|---|---|---|---|---|---|
| *Zamia vazquezii* | CR | Yes | P | 2 | 88 | 100% | 91% | 100% | 3 |
| *Zamia inermis* | CR | Yes | P | 1 | 50 | 100% | 74% | 100% | 1 |
| *Ceratozamia dominguezii* | EN | Yes | P | 5 | 1,318 | 100% | 87% | 100% | 3 |
| *Ceratozamia tenuis* | EN | Yes | P | 8 | 313 | 100% | 87% | 98% | 14 |
| *Ceratozamia morettii* | EN | Yes | P | 4 | 175 | 100% | 57% | 100% | 9 |
| *Ceratozamia mexicana* | CR | Yes | A | 6 | 599 | 100% | 82% | 100% | 12 |
| *Zamia furfuracea* | EN | Yes | P | 4 | 1,998 | 100% | 56% | 84% | 10 |
| *Ceratozamia brevifrons* | EN | Yes | A | 8 | 693 | 100% | 89% | 100% | 6 |
| *Ceratozamia decumbens* | EN | Yes | P | 4 | 495 | 100% | 63% | 70% | 14 |
| *Ceratozamia subroseophylla* | EN | Yes | P | 8 | 1,365 | 100% | 87% | 60% | 5 |
| *Ceratozamia delucana* | EN | Yes | A | 5 | 1,050 | 100% | 45% | 100% | 7 |
| *Ceratozamia huastecorum* | CR | Yes | A | 1 | 34 | 100% | 22% | 0% | 4 |
| *Ceratozamia miqueliana* | EN | No | P | 9 | 7,065 | 94% | 89% | 59% | 15 |
| *Ceratozamia euryphyllidia* | EN | No | P | 5 | 1,387 | 63% | 64% | 76% | 4 |
| *Dioon spinulosum* | EN | No | P | 3 | 2,110 | 22% | 53% | 98% | 9 |
| *Zamia purpurea* | CR | No | P | 14 | 4,696 | 70% | 47% | 95% | 9 |
| *Zamia splendens* | EN | No | P | 3 | 1,200 | 9% | 55% | 93% | 64 |
| *Ceratozamia fuscoviridis* | EN | No | A | 12 | 4,375 | 36% | 53% | 99% | 16 |
| *Ceratozamia totonacorum* | VU | No | A | 9 | 2,823 | 17% | 69% | 100% | 18 |
| *Zamia loddigesii* | NT | No | A | 15 | 130,031 | 83% | 54% | 99% | 201 |
| *Dioon edule* | NT | No | P | 18 | 97,472 | 9% | 14% | 84% | 74 |

was selected because it allows for using categorical and numerical variables, offering greater flexibility than Principal Component Analysis. The variables IUCN, endemism, and NOM-059 were treated as categorical (using their original values).

I clustered the species, using the same variables as in the MFA analysis, to identify the species with a higher risk of extinction. I applied hierarchical clustering with Gower's distance to determine the optimal number of clusters, accommodating numerical and categorical variables. I used Ward's method for clustering, as it minimises within-cluster variance. I performed a bootstrap resampling procedure with 5000 iterations using the *pvclust* package to ensure robust cluster identification. Clusters were validated by evaluating the Approximately Unbiased (AU) *p*-values, retaining only those with an AU ≥ 95%. Based on these results, the clusters were identified and visualised within the biplot of the MFA.

Finally, the CPI was calculated by summing all normalised variables and, where applicable, their inverses: normalised IUCN status, endemism to Veracruz, normalised NOM-059 status, inverse normalised number of localities, inverse normalised EOO, percentage of distribution in Veracruz, percentage of habitat loss, percentage of distribution outside of PA, and inverse normalised number of municipalities (Table 2). To ensure the reliability and accuracy of CPI data, I selected variables based on expert consultation with the Cycad Conservation Group in Veracruz, which includes the experts mentioned in acknowledgements. Most data were sourced from established databases like the IUCN and NOM;

gaps were supplemented from available literature. Robust spatial methods were employed to maintain data integrity and minimise potential biases.

I analysed the relationship between the CPI of all species and within the genera *Ceratozamia* and *Zamia* (*Dioon* was excluded in this latter analysis due to having only two species) and the longitude and latitude of the centroid of each species' polygon using general linear models (GLM) with a gamma error distribution, and the link = "log" setting. Additionally, I incorporated second-degree polynomial terms for the independent variables to capture potential quadratic relationships and model non-linear effects in the analysis (James et al., 2021).

Using the distribution maps, I conducted a conservation gap analysis for Zamiaceae species in Veracruz. First, all distribution polygons were rasterised using the *rasterize* function from the *raster* package (Hijmans, 2020) with a resolution of 1 km². Next, the maps were summed to obtain the richness of Zamiaceae species in Veracruz. I overlaid federal, state, and AVDC PA layers to visualise areas of species richness outside the current PA network. I conducted all analyses in this study using the R software (R Core Team, 2023).

### Knowledge shortfalls

To identify information gaps for each species, which directly impact conservation policies and actions, I conducted a species-specific information search using the IUCN assessments to obtain population

**Table 2.** Values used to calculate the Conservation Prioritization Index (CPI) for cycad species in Veracruz, Mexico, which include IUCN (normalised IUCN risk category), Endemic (1 for endemic, 0 for non-endemic), NOM-059 (normalised risk category under Mexican law NOM-059-SEMARNAT-2010), Locations (inverse normalised number of localities), EOO (inverse normalised Extent of Occurrence), Area Veracruz (percentage of the species' distribution within Veracruz), Habitat Loss (percentage of habitat loss), Out PA (percentage of the species' distribution outside Protected Areas), Municipalities (inverse normalised number of municipalities), and CPI (calculated as the sum of all standardised values)

| Species | IUCN | Endemic | NOM-059 | Locations | EOO | Area Veracruz | Habitat loss | Out PA | Municipalities | CPI |
|---|---|---|---|---|---|---|---|---|---|---|
| *Zamia vazquezii* | 1.0 | 1 | 1.0 | 0.94 | 1.0 | 1.0 | 0.9 | 1.0 | 1.0 | 8.837 |
| *Zamia inermis* | 1.0 | 1 | 1.0 | 1.00 | 1.0 | 1.0 | 0.7 | 1.0 | 1.0 | 8.742 |
| *Ceratozamia dominguezii* | 0.8 | 1 | 1.0 | 0.76 | 1.0 | 1.0 | 0.9 | 1.0 | 1.0 | 8.354 |
| *Ceratozamia tenuis* | 0.8 | 1 | 1.0 | 0.61 | 1.0 | 1.0 | 0.9 | 1.0 | 0.9 | 8.139 |
| *Ceratozamia morettii* | 0.8 | 1 | 1.0 | 0.85 | 1.0 | 1.0 | 0.6 | 1.0 | 1.0 | 8.125 |
| *Ceratozamia mexicana* | 1.0 | 1 | 0.7 | 0.70 | 1.0 | 1.0 | 0.8 | 1.0 | 0.9 | 8.119 |
| *Zamia furfuracea* | 0.8 | 1 | 1.0 | 0.82 | 1.0 | 1.0 | 0.6 | 0.8 | 1.0 | 7.905 |
| *Ceratozamia brevifrons* | 0.8 | 1 | 0.7 | 0.58 | 1.0 | 1.0 | 0.9 | 1.0 | 1.0 | 7.854 |
| *Ceratozamia decumbens* | 0.8 | 1 | 1.0 | 0.82 | 1.0 | 1.0 | 0.6 | 0.7 | 0.9 | 7.833 |
| *Ceratozamia subroseophylla* | 0.8 | 1 | 1.0 | 0.61 | 1.0 | 1.0 | 0.9 | 0.6 | 1.0 | 7.793 |
| *Ceratozamia delucana* | 0.8 | 1 | 0.7 | 0.79 | 1.0 | 1.0 | 0.4 | 1.0 | 1.0 | 7.616 |
| *Ceratozamia huastecorum* | 1.0 | 1 | 0.7 | 1.00 | 1.0 | 1.0 | 0.2 | 0.0 | 1.0 | 6.875 |
| *Ceratozamia miqueliana* | 0.8 | 0 | 1.0 | 0.52 | 0.9 | 0.9 | 0.9 | 0.6 | 0.9 | 6.569 |
| *Ceratozamia euryphyllidia* | 0.8 | 0 | 1.0 | 0.76 | 1.0 | 0.6 | 0.6 | 0.8 | 1.0 | 6.507 |
| *Dioon spinulosum* | 0.8 | 0 | 1.0 | 0.88 | 1.0 | 0.2 | 0.5 | 1.0 | 1.0 | 6.300 |
| *Zamia purpurea* | 1.0 | 0 | 1.0 | 0.21 | 1.0 | 0.7 | 0.5 | 1.0 | 1.0 | 6.255 |
| *Zamia splendens* | 0.8 | 0 | 1.0 | 0.88 | 1.0 | 0.1 | 0.6 | 0.9 | 0.7 | 5.884 |
| *Ceratozamia fuscoviridis* | 0.8 | 0 | 0.7 | 0.36 | 1.0 | 0.4 | 0.5 | 1.0 | 0.9 | 5.546 |
| *Ceratozamia totonacorum* | 0.5 | 0 | 0.7 | 0.52 | 1.0 | 0.2 | 0.7 | 1.0 | 0.9 | 5.432 |
| *Zamia loddigesii* | 0.3 | 0 | 0.7 | 0.15 | 0.0 | 0.8 | 0.5 | 1.0 | 0.0 | 3.425 |
| *Dioon edule* | 0.3 | 0 | 1.0 | 0.00 | 0.3 | 0.1 | 0.1 | 0.8 | 0.6 | 3.205 |

data. IUCN data were utilised due to the scarcity of population studies, which exist for only a few species and specific localities rather than their entire range. However, when available and relevant, these population studies were also incorporated into the analysis. Also, I looked in GenBank to get the number of genetic accessions and Google Scholar to determine the number of scientific papers available. Within Google Scholar, I used specific search criteria (see Supplementary Text S2) to determine if at least one study related to the population ecology of each species. Finally, through a database constructed in collaboration with the cycad pollination expert William Tang (unpublished data), I provide information on species for which the pollinator is already known. To rank species based on available information, I sum the number of fields with at least some information.

## Results

The correlation matrix for CPI variables showed that conservation status (IUCN classification) strongly correlates with inverse EOO ($r = 0.79$) and moderately with the inverse number of municipalities

($r = 0.72$; Supplementary Figure S3). It also highlights a strong positive correlation between endemism and the area covered within Veracruz ($r = 0.80$). Additionally, there was a moderate correlation between habitat loss percentage and EOO ($r = 0.40$).

In the MFA, the first dimension had the highest eigenvalue of 1.89, explaining 33.83 % of the total variance, while the second dimension had an eigenvalue of 0.88 (15.73 %). Together, the first two dimensions accounted for 49.56 % of the variance. Variables like the inverse number of localities, inverse number of municipalities, percentage of area in Veracruz, and habitat loss had substantial loadings on the first dimension, indicating a gradient of extinction risk correlated with the IUCN Critical Risk level and endemic species (Figure 1). The hierarchical cluster analysis revealed four species clusters, with eight species grouped in the high extinction risk cluster (Figure 1, group 3): *Z. vazquezii, Z. inermis, C. decumbens, Z. furfuracea, C. morettii, C. subroseophylla, C. tenuis, C. dominguezii.*

*Z. vazquezii* had the highest CPI score (8.837), followed by *Z. inermis* (8.742). *C. dominguezii, C. tenuis,* and *C. morettii* also had

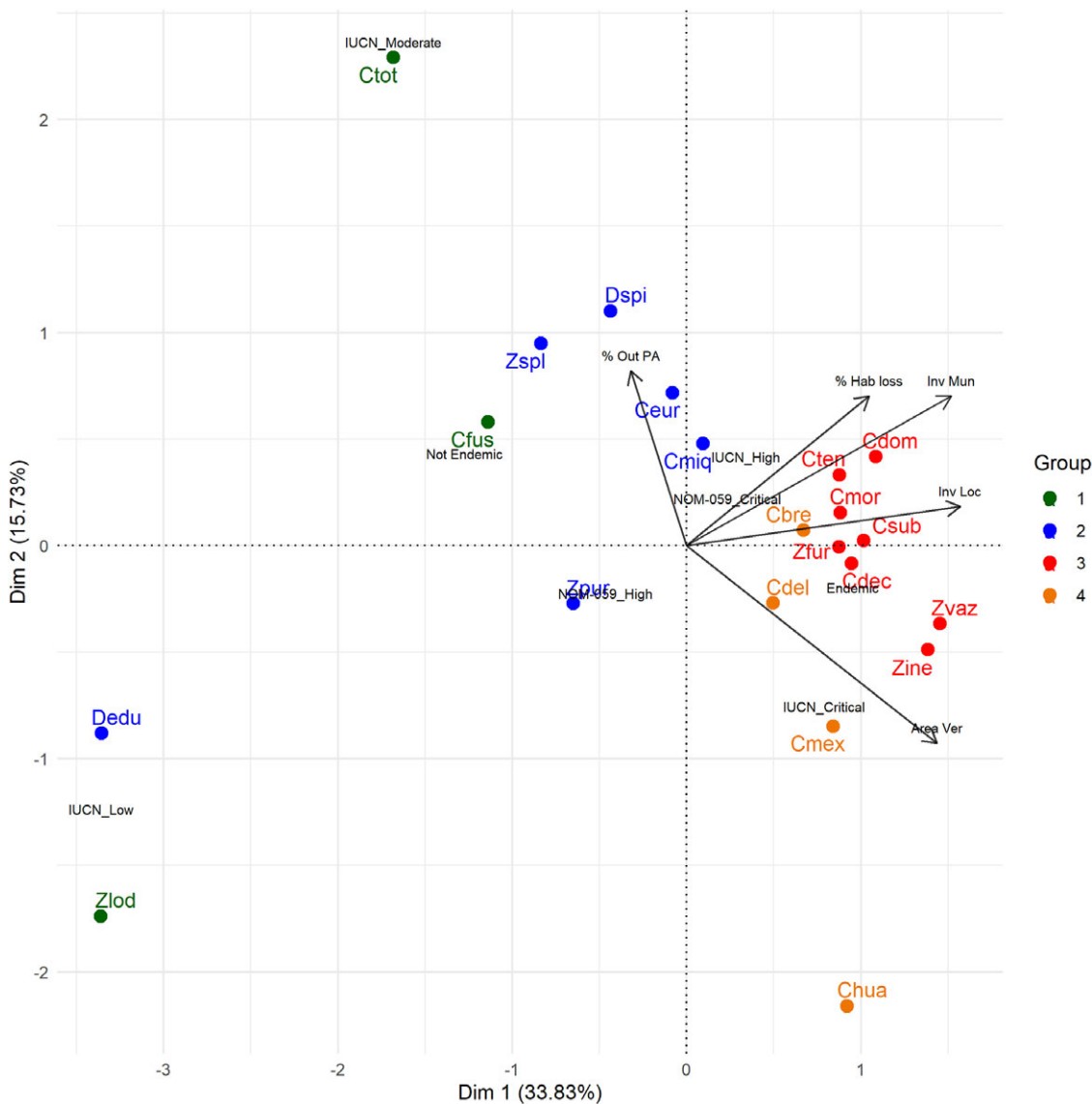

**Figure 1.** Biplot illustrating species ordination and key variable influences via multiple-factor analysis. The four groups of species, determined through hierarchical clustering with Gower's distance and Ward's method, are represented by different colours, highlighting distinct patterns about the influencing variables.

high scores (8.354, 8.139, and 8.125), indicating elevated conservation significance (Table 2). In contrast, *D. edule* and *Z. loddigesii* received lower scores (3.205 and 3.258), suggesting low conservation urgency (Table 2). Significant relationships were found between CPI and longitude ($p = 0.003$, $D^2 = 0.319$; Figure 2A), CPI and latitude ($p = 0.006$, $D^2 = 0.314$; Figure 2B) for all species, and between longitude and CPI for *Ceratozamia* ($p = 0.002$, $D^2 = 0.652$; Figure 3).

Southern Veracruz, particularly Uxpanapa, has a high Zamiaceae richness with four species: *C. euryphyllidia*, *C. dominguezii*, *Z. purpurea*, and *Z. loddigesii*. Other richness peaks occur in the Sierra Madre Oriental (Sierra de Chiconquiaco and Huatusco) and Southeast Veracruz (Santiago Tuxtla), hosting three species in each area (Figure 4). A conservation gap exists throughout Veracruz Zamiaceae distribution (Figure 4), lacking PA sites that safeguard areas with the highest species richness or those at significant risk of extinction according to the CPI.

Among the studied species, 52 % exhibited comprehensive individual data per IUCN, with counts ranging from 26 (*Z. vazquezii*) to 42,450 (*D. edule*; Table 3). GenBank analysis shows that 95 % possess at least one accession entry; *Z. vazquezii* has the most (11,558), followed by *Z. furfuracea* (1,132 entries; Table 3), while *C. dominguezii* lacks genetic information. All species have at least ten scholarly studies on Google Scholar (Table 3). *D. edule* leads with 1,240 studies, and *Z. furfuracea* and *C. mexicana* have 868 and 727 studies, respectively. On the contrary, several species exhibit a more limited scholarly presence, with *C. totonacorum*, *C. delucana*, and *C. subroseophylla* featuring 11, 11, and 14 studies, respectively.

Regarding studies on population ecology aspects such as demography, individual counts, population dynamics, and population genetics, there is only information for 38% of the studied cycad species (Table 3). Among the studied species, information is

available regarding the pollinators for 52% (Table 3). Based on the cumulative scores from individual criteria, the assessment of species knowledge reveals varying levels of available data (Table 3). *Z. inermis* and *C. miqueliana* emerge as the species with the most comprehensive information. On the other end of the spectrum, *C. dominguezii* has the lowest score of 1, indicating limited available data across the assessed criteria.

## Discussion

In this study, I evaluated the extinction risk of cycad species in Veracruz using new and existing data. VU species were identified based on this assessment. The Mexican NOM-059 and the IUCN assessments are not correlated because they use different criteria for assigning extinction risk (Frías-Alvarez et al., 2010) and should be seen as complementary. Similar findings were observed in Mexican amphibians, where government listings placed species in lower-risk categories than the IUCN (Frías-Alvarez et al., 2010). Integrating multiple data sources enhances the efficiency and robustness of species extinction risk assessments (Böhm et al., 2016). Our analysis shows that the two systems are not strongly correlated, and their orthogonal nature is reflected in the ordination analysis. This lack of correlation underscores the importance of using complementary criteria, such as the CPI, to bridge the gaps between international and national conservation frameworks.

In Veracruz, deforestation and land-use change are the main threats to cycad populations (Gómez-Díaz et al., 2024), making canopy height a useful indicator of healthy populations. While habitat degradation was not directly measured, satellite lidar imaging of vegetation height and loss of vegetation cover served as indirect indicators (Potapov et al., 2021). A similar study has

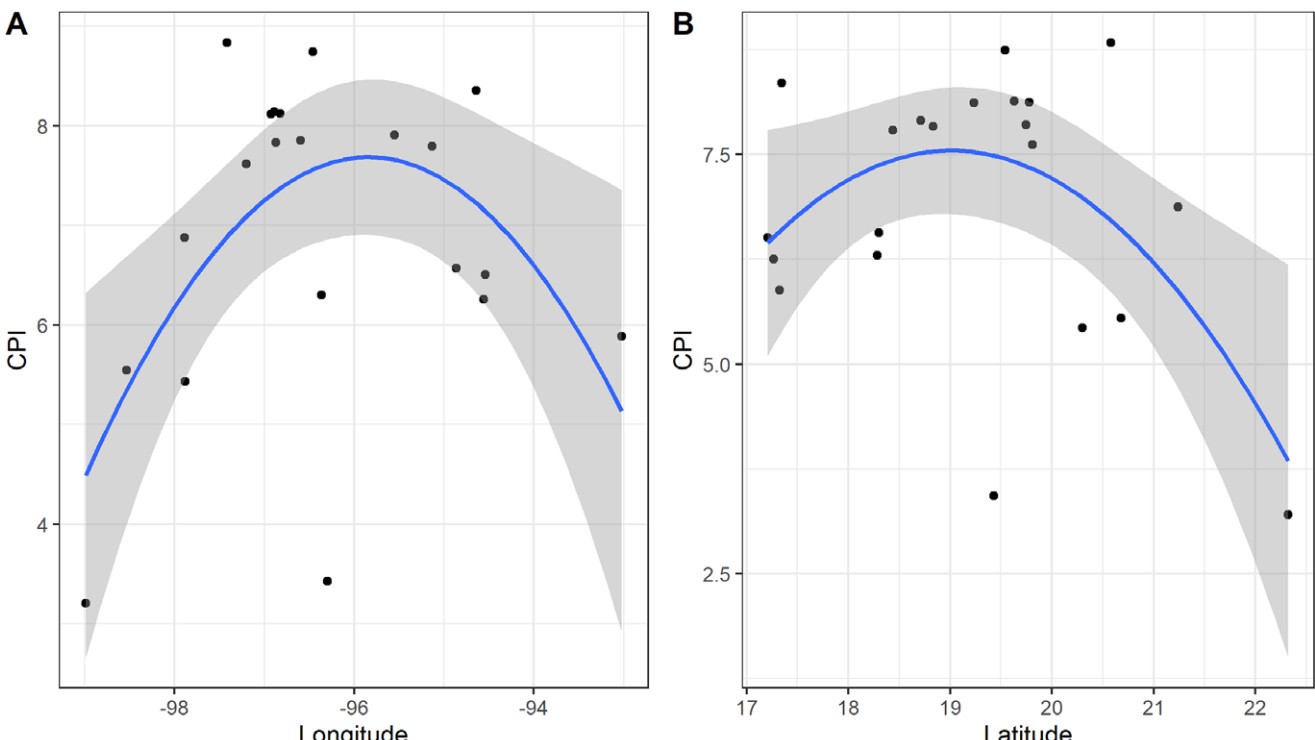

**Figure 2.** Relation between the latitude (A), longitude (B) and the Conservation Priority Index across all species of Zamiaceae found in Veracruz.

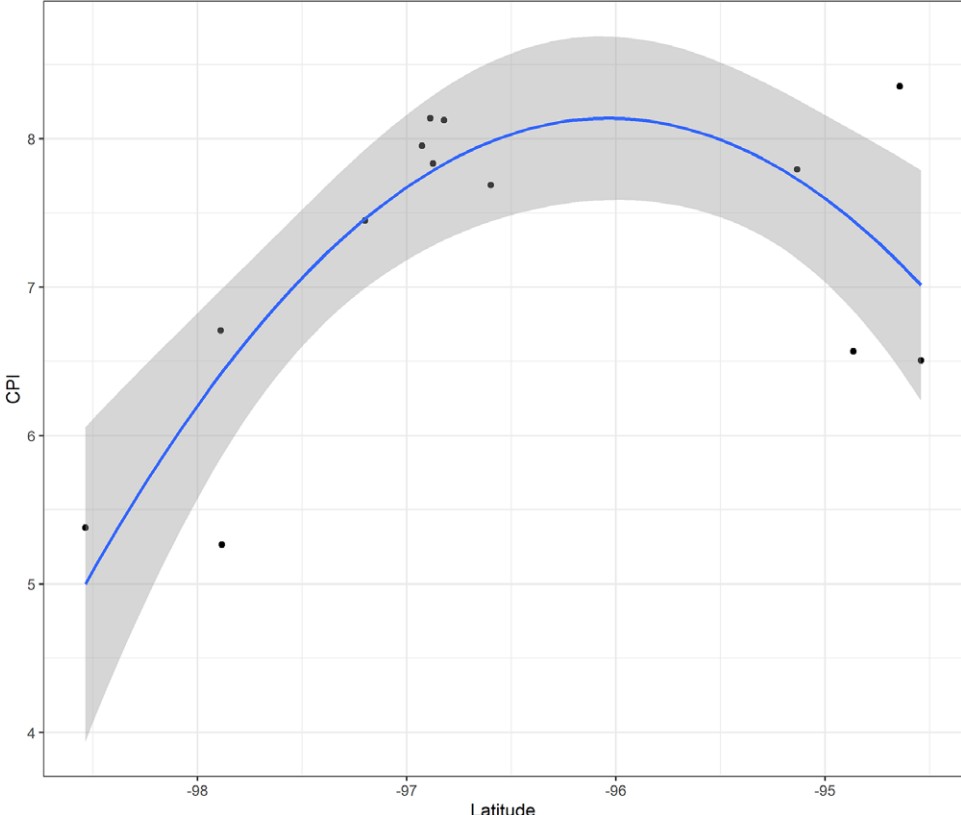

**Figure 3.** Relation between the Conservation Priority Index and the longitude of the genera *Ceratozamia* found in Veracruz.

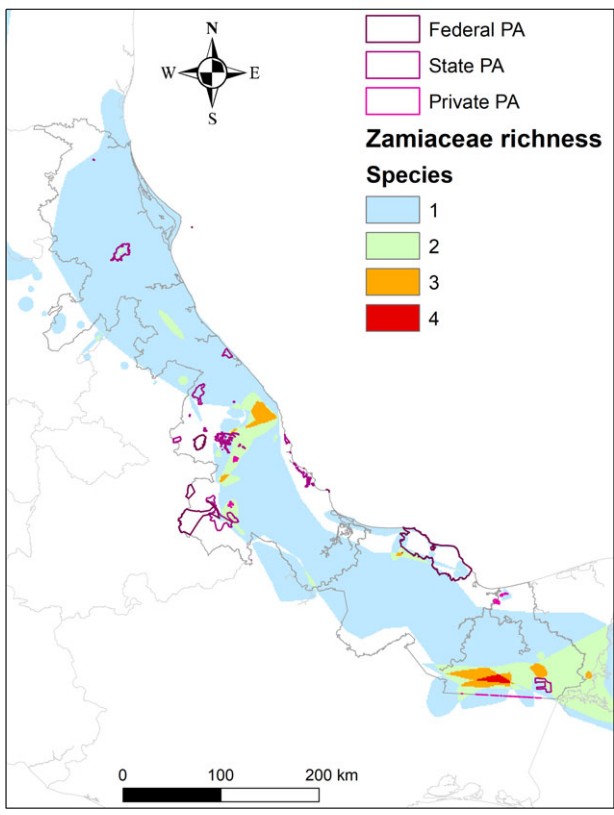

**Figure 4.** Species richness of Zamiaceae and the current network of Protected Areas in Veracruz.

successfully used this method (Carvajal-Hernández and Gómez-Díaz, 2024). However, the CPI does not currently incorporate other critical environmental variables, such as climate, geology, and vegetation type, which could significantly influence species distributions and extinction risk. For example, climate variables like temperature and precipitation patterns may affect cycad survival and reproduction, while geological factors could determine soil suitability and water availability (Álvarez-Yépiz et al., 2011). The exclusion of these variables may limit the CPI's ability to fully capture the ecological complexity of cycad habitats, potentially altering risk rankings if they are included in future analyses.

While canopy height data provides a useful approximation of habitat suitability by identifying relatively undisturbed areas, it is not a comprehensive measure of habitat quality. Additional data, including fine-scale environmental variables, are needed to understand better the full impact of habitat degradation on cycad populations (Lopez-Gallego and O'Neil, 2010). Incorporating these factors would require advanced modelling techniques, such as species distribution models, which are beyond this study's scope but important for future research (Swart et al., 2018; Carvajal-Hernández and Gómez-Díaz, 2024). Updating the CPI to include these variables would enhance its accuracy and provide a more robust foundation for conservation strategies. Recognising and addressing the factors that contribute to species vulnerability is essential for designing effective conservation measures (Cardona, 2013).

Three variables correlate with the IUCN assessment, suggesting they can act as indicators when comprehensive IUCN data is lacking. Another study highlighted range size as a primary predictor of extinction risk, particularly in reptiles assessed under range-based

**Table 3.** Knowledge shortfalls for the species of Zamiaceae of Veracruz, including number of individuals (ind.), genetic information (GB), studies (Google Scholar), population ecology, pollinators and the sum of the fields with information

| Species | Ind. | GB | Google Scholar | Population ecology | Pollinators | Sum |
|---|---|---|---|---|---|---|
| *C. miqueliana* | 1,013 | 33 | 151 | 1 | 1 | 5 |
| *Z. inermis* | 654[1] | 61 | 91 | 1 | 1 | 5 |
| *D. edule* | 42,450[2] | 169 | 1,240 | 1 | | 4 |
| *D. spinulosum* | 15,000 | 112 | 447 | 1 | | 4 |
| *C. brevifrons* | 4,000 | 4 | 45 | | 1 | 4 |
| *C. fuscoviridis* | 1,636 | 18 | 90 | 1 | | 4 |
| *C. morettii* | 1,000 | 12 | 34 | | 1 | 4 |
| *C. euryphyllidia* | 340 | 9 | 83 | | 1 | 4 |
| *C. mexicana* | | 51 | 727 | 1 | 1 | 4 |
| *Z. furfuracea* | | 1,132 | 868 | 1 | 1 | 4 |
| *Z. purpurea* | 2,000 | 18 | 54 | | | 3 |
| *C. decumbens* | 1,000 | 9 | 26 | | | 3 |
| *Z. vazquezii* | 26 | 11,558 | 79 | | | 3 |
| *Z. loddigesii* | | 102 | 309 | 1 | | 3 |
| *C. huastecorum* | | 9 | 20 | | 1 | 3 |
| *C. subroseophylla* | | 2 | 14 | | 1 | 3 |
| *C. delucana* | | 2 | 11 | | 1 | 3 |
| *C. tenuis* | | 1 | 38 | | 1 | 3 |
| *Z. splendens* | | 22 | 48 | 1 | | 3 |
| *C. totonacorum* | | 2 | 11 | | | 2 |
| *C. dominguezii* | | | 13 | | | 1 |

[1]Based on the study of Octavio-Aguilar *et al.* (Octavio-Aguilar et al., 2017).
[2]Based on IUCN data, it is mentioned that currently, there are at least 30% of the 141,500 plants recorded between 1998 and 2005.

IUCN criteria (Böhm et al., 2016). Species with restricted ranges and specialised habitats accessible to humans, like cycads, are more likely to face early extinction (Böhm et al., 2016). Geographic range and distribution are critical in assessing conservation status, especially for species with limited ranges vulnerable to extinction threats (Böhm et al., 2016). While correlations offer useful insights, they must be interpreted cautiously since correlation does not imply causation, and factors such as unaccounted environmental variables or human pressures could influence observed relationships (Loney and Nagelkerke, 2014).

Two species, *Z. vazquezii* and *Z. inermis*, face the highest extinction risk among Veracruz cycads. *Z. vazquezii* has a limited presence in northern Veracruz and is severely threatened by habitat loss and over-collection for ornamental use (Bösenberg, 2023). It is known from only two locations with at most 50 wild plants. Deforestation has caused ~91 % of its habitat loss, leading to severe population fragmentation. The population trend for this species is decreasing, with only 2–49 mature individuals observed (Bösenberg, 2023). Education, awareness programs, and long-term monitoring are crucial for conserving this critically endangered species.

*Z. inermis* has only one documented natural population with a declining growth rate ($\lambda = 0.963 \pm 0.011$), highlighting its precarious status (Octavio-Aguilar et al., 2017). Few juveniles, no seedling survival in the field, and low viable seeds per female cone underscore the population's challenges (Octavio-Aguilar et al., 2017). Local inhabitants aid conservation by cultivating the species in backyard nurseries, but *Z. inermis* relies solely on adult plants, increasing its risk (Octavio-Aguilar et al., 2017). While *ex-situ* propagation is necessary, it must be carefully managed to avoid issues like disturbing pollinator breeding sites by removing male cones for artificial pollination (Swart, 2019). Thus, careful planning and monitoring of propagation efforts are essential to prevent further decline of wild populations. *Z. vazquezii* and *Z. inermis* with *Z. fischeri* form the Fischeri clade, sister to all other mainland *Zamia* species (Calonje et al., 2019). The imminent extinction threatens an entire evolutionary clade of cycads. The interconnectedness within the Fischeri clade highlights the urgent need for conservation efforts to protect individual species and their collective lineage.

The next priority species are from the genus *Ceratozamia*: *C. dominguezii*, *C. tenuis*, *C. morettii*, and *C. mexicana*. *C. dominguezii*, recently described, is found in five populations in Uxpanapa, Veracruz (Pérez-Farrera et al., 2021). However, the Uxpanapa forest has been severely impacted, with ~87 % of its habitat transformed (Hernández et al., 2013). Urgent conservation efforts are needed to protect these species in their natural habitat and establish *ex-situ* collections for their safeguarding (Pérez-Farrera et al., 2021). *C. tenuis* and *C. morettii* inhabit central Veracruz southern cloud forests in the Sierra de Chiconquiaco. Distribution simulations since the last glacial maximum indicate a decline for both species (Gómez-Díaz et al., 2024). Future climate change is projected to reduce their ranges by 83% and 88 %, respectively, nearly

driving them to extinction except in limited southern mountainous areas (Gómez-Díaz et al., 2024). Habitat suitability models show that droughts will negatively impact these cycads, which thrive in humid and temperate conditions (Gómez-Díaz et al., 2024).

*C. mexicana* is a slow-growing, long-lived species found in the understory of central Veracruz Mountain cloud forests. Agricultural habitat destruction has led to a population decline and isolation, with ~82 % of its habitat transformed. Populations in disturbed sites have experienced recent bottlenecks, resulting in a loss of genetic diversity (Rivera-Fernández, 2012). Corrective measures are essential to prevent further forest degradation and genetic loss. The Mexicana subclade includes *C. mexicana* and *C. tenuis* (Gutiérrez-Ortega et al., 2024). Like the Fischeri clade, the high extinction risk of these two species may cause the extinction of an entire subclade within the *Ceratozamia* genus.

Selecting variables for the proposed CPI may involve subjectivity, but including expert and stakeholder input ensures a comprehensive and balanced approach. Variables such as habitat loss and species distribution are dynamic and can change over time (Ney-Nifle and Mangel, 2000), necessitating periodic CPI updates every five years to incorporate new scientific knowledge and adapt conservation priorities. Waiting for perfect and complete data risks missing critical conservation opportunities (Cronk, 2016; Swart, 2019). For example, although Veracruz cycads have been studied since the 1970s, identifying the most at-risk species has only recently been coordinated. Further delays could hinder the timely protection of this group.

While the CPI provides a comprehensive framework for assessing the extinction risk of Veracruz cycads, it is crucial to recognise the limitations of incomplete data for many species (Tafirei, 2016). The CPI relies on ecological, geographical, and legal criteria. Still, the lack of complete datasets – particularly genetic information, population sizes, and field-verified distribution data – may introduce biases in the risk assessments. Currently, missing attributes are omitted from the summation process, which could lead to underestimating extinction risks for certain species. For instance, the absence of genetic data limits our ability to evaluate the resilience and adaptive potential of species, which are critical factors for long-term survival (Mable, 2019). Similarly, incomplete population data may lead to underestimating the extinction risk for species with small or declining populations (Purvis et al., 2000).

This absence of data arises from several historical, logistical, and biological challenges. First, cycads are often located in remote or inaccessible areas, making field surveys and data collection labour-intensive and costly (Carvajal-Hernández and Gómez-Díaz, 2024). Second, the long life cycles and slow reproductive rates of cycads complicate population monitoring, as changes in population size or structure may not be immediately noticeable (Raimondo and Donaldson, 2003). Third, genetic studies on cycads in Veracruz are still scarce, partly due to genomic research's technical and financial limitations. These factors collectively contribute to significant distribution, population, and genetic data gaps.

To address these limitations, future research should prioritise targeted field verification, genetic sampling, and stochastic resampling methods to quantify uncertainty and improve the accuracy of risk assessments (Fordham et al., 2016). Additionally, contrasting the IUCN and MER criteria more explicitly in future iterations of the CPI would improve its robustness. Despite these challenges, the CPI represents a significant initial effort to prioritise conservation actions for Veracruz cycads. By iteratively updating the index with new data and incorporating genetic and ecological insights, the CPI

can evolve into a more precise tool for guiding conservation efforts. This proactive approach, while imperfect, is vital to prevent further declines in cycad populations and ensure the timely protection of this ecologically and evolutionarily significant group.

Proactive conservation is more effective and cost-efficient than reactive efforts to recover declining populations. Acting before species reach critical threat levels allows for the easier and cheaper implementation of measures, increasing the likelihood of success (Possingham et al., 2002; McCarthy et al., 2012). In contrast, restoring populations after significant declines requires larger investments and faces greater biological and socioeconomic challenges (Dobson et al., 1997). For cycads in Veracruz, proactive conservation is essential to prevent drastic population declines. Implementing measures early is more efficient and effective, as seen in other species (Farrera and Vovides, 2004).

The Gulf Coast plains likely influence the relationship between longitude and the CPI of Ceratozamia species in Veracruz. Coastal populations are more exposed to natural disturbances and human activities such as urban development and agriculture (Donaldson, 2003; Martinez et al., 2017), increasing their vulnerability to habitat loss and ecosystem changes. To fully understand this phenomenon, specific studies should analyse the environmental conditions, local threats, and ecological characteristics of each *Ceratozamia* species across different locations in the region.

The first dimension of the MFA shows a strong positive contribution from factors like IUCN status and the inverse number of localities and municipalities. This suggests a measure of extinction risk, emphasising the importance of conserving areas with high diversity and endemism (Pollock et al., 2017). Conversely, the second dimension reflects the positive influence of the extent of area inside Veracruz, NOM-059 status and the percentage of area outside PA, indicating that areas with greater habitat diversity are also associated with higher biodiversity loss (Mantyka-pringle et al., 2012).

The highest cycad diversity in Veracruz is found in Uxpanapa for two main reasons. First, as tropical gymnosperms, cycads thrive in the state's best-preserved rainforest (Sandoval Mendoza et al., 2006; Hernández et al., 2013). Second, Uxpanapa is a centre of high diversity and endemism due to factors like Pleistocene refuge, geographical isolation, karstic geology, and high precipitation (Hernández et al., 2013; Molina-Paniagua et al., 2023). Another area with high Zamiaceae richness in Veracruz is the confluence region between the Sierra Madre Oriental and the Trans-Mexican Volcanic Belt. This area is highly biodiverse because it lies at the intersection of two biogeographic regions, has high humidity, and features Veracruz's most diverse vegetation type: cloud forest (Gómez Díaz et al., 2023; Gómez-Díaz et al., 2024).

The existing PA network includes no areas with high Zamiaceae diversity in Veracruz. To conserve this highly threatened group, expanding the PA network or implementing alternative protection strategies is necessary (Gómez Díaz et al., 2023). One approach is to identify and designate Key Biodiversity Areas (KBAs), which focus on species not adequately represented in PA (Eken et al., 2004). The CPI developed in this study can be a valuable tool to pinpoint critical sites requiring immediate conservation based on threat levels, ecological importance, and species vulnerability. Furthermore, the CPI could be integrated into local, national, and international conservation policies to enhance the protection of cycads in Veracruz. Aligning the CPI with frameworks like KBAs would better define conservation priorities, enable more effective management decisions, and foster collaboration among stakeholders such as conservation organisations,

government agencies, and local communities. This integration would address gaps in the PA network and support broader biodiversity conservation strategies.

In this evaluation, three *Zamia* species – *Z. vazquezii*, *Z. inermis*, and *Z. furfuracea* – are at high risk of extinction. However, CITES currently list all *Zamia* species except *Z. restrepoi* under Appendix II, which includes species requiring trade control but not necessarily threatened with extinction (CITES, 2023). This may not accurately reflect the urgent conservation needs of these species (Sadiki, 2021). I recommend reclassifying the high-risk *Zamia* species to Appendix I of CITES to ensure enhanced protection. Furthermore, the CPI from this study can be integrated into CITES Non-Detriment Findings (NDF) to evaluate the three *Zamia* species. Elements such as threat level, ecological importance, genetic diversity, rarity, and human impact offer key insights into their conservation needs and vulnerabilities. Incorporating these factors into the NDF process would improve the accuracy of conservation assessments and inform trade regulations. This integration would also promote stakeholder engagement and collaboration among scientific and conservation organisations, government agencies, and local communities, leading to more effective conservation strategies for *Zamia* species.

There is a critical need for enhanced data collection on cycad species in Veracruz, reflecting broader biodiversity challenges and knowledge gaps (Hortal et al., 2015; Nori et al., 2023). This study found that only 10% of the studied species have complete data, with data heterogeneity leading to potential inaccuracies. I supplemented missing information using proposed methods and standardised all variables to address these gaps. This deficit hampers our understanding of cycad ecology, conservation, and population dynamics. Improving data quality and breadth is essential for effective conservation strategies and comprehensive ecological knowledge of these species (Nori et al., 2023).

Drawing upon the scientific findings and acknowledging the social dimensions of conservation, I propose specific, actionable measures to protect and restore cycad populations in Veracruz. For the top-ranked at-risk species, such as *Z. vazquezii* and *Z. inermis*, urgent interventions should include targeted habitat restoration in areas with high deforestation rates and stricter enforcement of protected area boundaries to prevent illegal land use and poaching. Additionally, KBAs should be designated or expanded to protect critical habitats, particularly for species with restricted distributions or high habitat loss. These actions should be complemented by community-based conservation programs that engage local stakeholders in habitat monitoring and restoration efforts, fostering a sense of ownership and responsibility.

A multi-faceted approach is essential to ensure the long-term survival of Veracruz cycads. This includes: (1) habitat protection and restoration: identifying and prioritising critical habitats, including existing PA and regions with high potential for restoration; (2) monitoring and research: implementing long-term monitoring programs to track population trends, reproductive success, and the impacts of threats such as habitat loss and climate change; (3) conservation education and outreach: raising awareness among local communities, stakeholders, and the public about the ecological and cultural importance of cycads; (4) *in situ* and *ex-situ* conservation: establish *ex-situ* conservation measures, such as botanical gardens or seed banks, to safeguard genetic diversity and provide a backup for wild populations; (5) collaboration and partnerships: fostering collaboration among scientists, conservation organisations, governmental institutions, and local communities to develop coordinated conservation plans.

Policy recommendations should support these efforts, such as updating CITES listings for highly threatened species and integrating cycad conservation into regional and national biodiversity strategies. The proposed conservation measures aim to address the challenges cycad populations face in Veracruz by combining scientific knowledge, population data, and understanding social dynamics. These measures provide a comprehensive approach to protecting and restoring cycad populations, considering both the species' ecological requirements and the involvement of local communities and stakeholders in conservation efforts.

## Conclusions

In conclusion, this research has shed light on the extinction risk of cycad species in Veracruz, underscoring the urgency of incorporating this concern into conservation initiatives. The findings highlight the need for ongoing research and vigilant monitoring to guide the development of effective conservation policies and strategies. This study assessed the risk and proposed targeted conservation measures to safeguard and revitalise cycad populations in the region. These measures offer a holistic approach by addressing scientific and social dimensions, ensuring Veracruz cycads' enduring survival and conservation. This effort aligns with the broader objective of preserving biodiversity and promoting sustainable regional development. The findings of this study position Veracruz as a model for other states in Mexico and countries around the world to follow in terms of cycad conservation. Using the CPI to structure a conservation plan represents a novel and quantitative approach to conservation planning. Adopting this approach can significantly improve the effectiveness of conservation strategies globally, contributing to the preservation of threatened species more effectively and sustainably.

**Open peer review.**   To view the open peer review materials for this article, please visit http://doi.org/10.1017/ext.2025.5.

**Supplementary material.**   The supplementary material for this article can be found at http://doi.org/10.1017/ext.2025.5.

**Data availability statement.** Data available within the article or its supplementary materials.

**Acknowledgements.**   I thank Mario Vázquez Torres, Miguel Pérez Farrera, William Tang and César Isidro Carvajal Hernández for supporting the information provided for some of the studied species. Also, I thank Vanessa Handley and Tim Gregory for their support in developing this study. I employed ChatGPT 3.5, an AI tool, to refine language and translation tasks in the manuscript, enhancing clarity, coherence, and language proficiency in selected sections. This acknowledgement fulfils reporting requirements, emphasising that ChatGPT 3.5's role was limited to language refinement and did not influence the conceptualisation or interpretation of research findings. The ethical use of ChatGPT 3.5 adheres to guidelines, and any potential biases or competing interests arising from its use are transparently disclosed in the manuscript. William Tang reviewed the final English version to enhance language clarity and coherence without influencing the conceptualisation or interpretation of the research findings.

**Author contribution.**   JAGD conceived the idea, analysed it, and wrote the article.

**Financial support.**   I sincerely thank the Global Conservation Consortium for Cycads for their invaluable financial support, which played a crucial role in facilitating this research endeavour. I appreciate their commitment to advancing scientific knowledge and conservation efforts, underscoring the importance of collaborative initiatives in addressing challenges related to cycad extinction risk.

**Competing interests.**   The authors declare none.

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
