## [Reviewer Report]

The article proposes a model for a comprehensive approach to the risk of extinction of Veracruz cycads. The systematization of the information allows different bibliographic and methodological sources of office estimation to be combined, making it a relevant article for the magazine.

However, there are two substantial problems, the information available for the species is very heterogeneous, which generates gaps that underestimate the estimates in poorly studied species and, on the contrary, overestimates for the species with more information available.

The second problem is the multivariate approach. Better management of this type of tool is required to avoid violating assumptions, especially the nature of the data and multiple adjustment tests.

Finally, the writing requires substantial revision by a native English speaker and avoidance of superlatives.

I add an anonymous address where you can access comments on the writing.

https://drive.google.com/file/d/1Re3fVo_DQiR4qfXA-U0riZX2SnK6I8U7/view?usp=sharing

---

## [Reviewer Report]

Overview

The research paper attempts to provide a quantitative measure to rank the cycads of Veracruz according to their risk of extinction by using the Conservation Prioritization Index (CPI) method. This was done to prioritize species according to their conservation status to ensure that limited resources are allocated to the most vulnerable species in need of urgent conservation action.

Overall, it is an interesting and important study, but it can be improved with a more in-depth analysis of some variables used to develop the CPI.

Introduction

Line 47: Cycads were previously regarded as an ancient group of gymnosperms. Modern-day cycads are known to have recently diversified from a few ancestor species in the late Miocene (Crisp and Cook 2011; Nagalingum et al. 2011).

After line 73: It might be worth including the concept of extinction debt and cycads in the introduction (theoretical background). Cronk (2016) explored extinction lag times experienced by long-lived plants such as cycads, sometimes measured in many centuries. The slow-burning fuse of plant extinction is relevant to studies assessing the extinction risk of cycads. It is vital that conservation action is taken during the window period before the extinction process is complete, but at the same time, the cautionary approach should be applied to ensure that conservation action does not speed up lag time to extinction - see Type I, Type II and Type III conservation intervention types explored in Cronk (2016) Swart (2019 - PhD thesis).

Methodology

Variable selection: How did the researcher ensure that only reliable data/data of high quality was used to include in the CPI. Inaccurate/incomplete data could lead to biased or unreliable CPI scores, affecting the validity of conservation decisions. The choice of variables is also subjective; researchers/stakeholders may have varying opinions on which variables are most important. The research will be strengthened if experts and stakeholders are part of the variable selection process to ensure a comprehensive and inclusive approach. In addition, some variables may change over time; how does the CPI accommodate updates and revisions based on new scientific knowledge?

Line 146 & 162: Using records from the Global Biodiversity Information Facility (GBIF) in the research paper can be valuable but the accuracy and completeness of locality records can vary. Was the information used in the research paper scrutinized for accuracy by assessing the metadata associated with each record (information about the data source, collection methods, whether the point is georeferenced or not, etc)? Was the taxonomic identification of the locality record verified? If possible, field verification of records should be done to help validate accuracy and confirm species identifications.

Line 173: The identification of suitable habitat is based on one set of data (Global Canopy Height). This is simplistic given the many factors influencing habitat suitability for cycads. Were other environmental factors for the different species considered (geology, climate, vegetation type, etc.)? Gomez-Dıaz (2024) used bioclimatic data to determine the distribution of Ceratozamia morettii, C. brevifrons, and C. Tenuis and found that each species occupies a unique ecoregion and climatic niche. Similarly, Swart (2019) found that geology was an important environmental predictor in determining suitable habitat for Encephalartos latifrons (South African cycad). The researchers may want to expand on this to strengthen the results of the research and include other environmental factors that may be important to include. The resolution of the datasets used and the software used in the computation is also unclear.

Results

It is important to consider geographic range and distribution patterns when assessing species' conservation status and prioritizing conservation efforts, particularly for species with limited ranges that may be more vulnerable to extinction threats. While correlation is a useful statistical measure for quantifying relationships between variables, caution is necessary when interpreting correlations. Potential confounding factors, alternative explanations, and the need for further causal inference methods should be explored or mentioned in the text.

Discussion

Line 322: Reword? This sentence does not make sense. ...such as......change to .....as with......?

Line 357: As mentioned in the introduction, the type of conservation intervention is important and should be applied with caution. It was assumed that Encephalartos latifrons cycad had lost its regenerative potential, allowing landowners to propagate seedlings from wild parent plants but this comes with some serious risks and can further contribute to the decline of some populations e.g. harvesting male cones for pollen used in artificial pollination removes the breeding site for important weevil pollinators (see Swart PhD thesis for more information).

Line 421: .......are part of the current [PA] network......reword

Line 428: It may be worth mentioning how the CPI index developed can be incorporated into (CITES Non-detriment) NDF findings for the three Zamia species. Components of the CPI, such as threat level, ecological importance, genetic diversity, rarity and human impact, can be used to assess the species' overall conservation needs and vulnerabilities as part of the NDF process. This will also encourage stakeholder engagement and collaboration among scientific experts, conservation organizations, government agencies, local communities etc.

Discussion relating to Line 287: Key Biodiversity Areas (KBAs) are an important approach to filling conservation gaps for species not represented in PAs (Eken et al. 2004).

It may be worth exploring how the CPI fits in with and could be used in local, national and international conservation policy development improving the conservation of cycads in Veracruz.

References:

Crisp MD, Cook LG (2011) Cenozoic extinctions account for the low diversity of extant gymnosperms compared with angiosperms. New Phytol 192:997–1009

Nagalingum NS, Marshall CR, Quental, TB, Rai HS, Little DP and Mathews S (2011) Recent synchronous radiation of a living fossil. Science 334: 796 – 799

Cronk Q (2016) Plant extinctions take time. Science 353: 446–448

Swart, C., Donaldson, J. & Barker, N. Predicting the distribution of Encephalartos latifrons, a critically endangered cycad in South Africa. Biodivers Conserv 27, 1961–1980 (2018).

Swart C (2019) The conservation, ecology, and distribution of the Critically Endangered Encephalartos latifrons Lehm. (PhD thesis)

Line 564: Correct reference year: Gomez-Dıaz JA, Carvajal-Hernandez CI, Dattilo W (2024) Past, present and future in the geographical distribution of Mexican Tepezmaite cycads: Genus Ceratozamia. PLoS ONE 19(2): e0284007

Eken G, Bennun L, Brooks TM, Darwall W, Fishpool DC, Foster M, Knox D, Langhammer P, Matiku P, Radford E, Salaman P, Sechrest WES, Smith ML, Tordoff A (2004) Key Biodiversity Areas as site conservation targets. Bioscience 54: 1110–1118

---

## [Reviewer Report]

The work provides a comprehensive overview, but unfortunately, many species lack complete data, leading to bias and the need for field verification. Another issue is that the absence of data is not considered; instead, the program disregards the attribute when summing. Despite these challenges, it represents a significant effort and an acceptable initial approach. It paves the way for future research to confirm whether the model suits each species and fills the existing information gaps, particularly genetic data. This aspect should be included in the discussion, not only environmental stochasticity.

I also propose a contrast between risk classification models, as the IUCN criteria are much more extensive than the MER. Yet ignoring the value that missing data would have has serious shortcomings. There is also no averaging, stochastic calculation, or suggested resampling to assess temporal differences.

---

## [Editor Report]

I thank the authors for their comprehensive revision of the MS. Some further critical comments were offered by the original reviewer #1 that will require some consideration.

In short, the referee’s concerns about incomplete data and the absence of a clear accounting for missing information are legitimate, as is their suggestion that ignoring absent data potentially skews the results. A brief but direct acknowledgment of how missing attributes could bias the CPI, and a plan for iterative improvement as more data become available, would suffice in the revision. Their call to contrast IUCN and MER criteria more explicitly and to incorporate genetic data gaps into the discussion also seems valid. Incorporating a short paragraph acknowledging these deficiencies, explaining why they exist, and proposing future work (e.g., targeted field verification, genetic sampling, or stochastic resampling methods) would both satisfy these concerns and enhance the manuscript’s scientific rigor.

From my perspective as Handling Editor, a few other areas could still be strengthened. First, more explicit discussion should be included on the limitations and uncertainties inherent in the CPI method, especially given the data shortfalls. The text should clarify how missing or incomplete population and genetic data might bias the CPI and how these biases can be mitigated. Also, while the importance of habitat loss is underscored, some finer discussion is needed about why certain environmental variables (e.g., climate or geology) were excluded, and how their incorporation could alter the risk rankings. Finally, although the methods section is extremely detailed, but the reasoning behind each chosen variable in the CPI (and not others) could be more succinctly justified. Additionally, the emphasis on practical conservation outcomes and policy recommendations could be sharpened. While the manuscript does mention how findings might guide key biodiversity area designations or inform CITES listings, it could more directly map the results to concrete conservation interventions. For instance, specifying which management actions (e.g., targeted habitat restoration, stricter enforcement of protected area boundaries) might be most urgent for the top-ranked at-risk species would strengthen the impact and clarity of the discussion.

---

## [Editor Report]

I thank the author for the thorough response to the critiques of the R1 revision (from one reviewer in R2 and myself), and for clearly tracking these changes in the revised MS. After 2 rounds of review and revision, I am satisfied that the MS is now ready for publication.